# Research Progress on Heat Stress Response Mechanism and Control Measures in Medicinal Plants

**DOI:** 10.3390/ijms25168600

**Published:** 2024-08-07

**Authors:** Ziwei Zhu, Ying Bao, Yixi Yang, Qi Zhao, Rui Li

**Affiliations:** 1Engineering Research Center of Sichuan-Tibet Traditional Medicinal Plant, Chengdu University, Chengdu 610106, China; zhuziwei@cdu.edu.cn (Z.Z.); 13519647316@163.com (Y.B.); yangyixi1011@cdu.edu.cn (Y.Y.); 2Institute for Advanced Study, Chengdu University, Chengdu 610106, China; 3School of Food and Biological Engineering, Chengdu University, Chengdu 610106, China

**Keywords:** heat stress, medicinal plants, physiological changes, molecular mechanisms, control measures

## Abstract

Medicinal plants play a pivotal role in traditional medicine and modern pharmacology due to their various bioactive compounds. However, heat stress caused by climate change will seriously affect the survival and quality of medicinal plants. In this review, we update our understanding of the research progress on medicinal plants’ response mechanisms and control measures under heat stress over the last decade. This includes physiological changes, molecular mechanisms, and technical means to improve the heat tolerance of medicinal plants under heat stress. It provides a reference for cultivating heat-resistant varieties of medicinal plants and the rational utilization of control measures to improve the heat resistance of medicinal plants.

## 1. Introduction

Due to their therapeutic properties and healing potential, medicinal plants play a critical role in human health and well-being. However, as a consequence of global climate change, heat stress (HS) affects the growth, development, and distribution of medicinal plants. Medicinal plants are usually grown in specific and sometimes extreme environments, such as tropical rainforests, deserts, and mountains, which create their unique properties. Thus, medicinal plants, especially those in specialized habitats, face severe survival challenges under HS caused by climate change. Understanding the HS response mechanisms and implementing control measures in medicinal plants is crucial for ensuring sustainable cultivation and maximizing medicinal properties in these plants.

HS may cause a reduction in biomass and changes in biochemical content and composition, affecting the quality and safety of medicinal products [1]. Medicinal plants have developed complex tolerance mechanisms in response to HS. They protect cells against HS by accumulating osmotic substances, increasing antioxidant enzyme activities, and synthesizing antioxidants such as glutathione and ascorbic acid. Additionally, some medicinal plants promote the accumulation of secondary metabolites like aromatic amino acids and medicinal active ingredients in response to HS [1,2,3]. For example, in *Catharanthus roseus*, HS significantly decreased growth and biomass accumulation yet increased the accumulation of osmolytes and some secondary metabolites (e.g., tannins, terpenoids, and alkaloids), which may contribute to the maintenance of growth under stress conditions by improving the plant water potential and increasing reactive oxygen species (ROS) scavenging [4,5]. Increased accumulation of these secondary metabolites in medicinal plants may counteract the deleterious effects of HS.

The molecular mechanisms underlying plant responses to elevated temperatures have been extensively explored in *Arabidopsis* and crops [6]. Based on these studies, research on the response mechanisms to HS in medicinal plants has been conducted, with heat shock factors (HSFs) playing a central role. As transcription factors, HSFs can up-regulate a variety of HS-induced genes to activate heat shock responses, and they are also involved in an intricate protein–protein interaction network, which increases the complexity of their biological functions [7]. HS triggers some signaling molecules (e.g., Ca^2+^, ethylene) that induce the expression of various transcription factors (e.g., HSF, WRKY, MYB) in cells, causing a series of transcriptional reprogramming events. The induced transcription factors activate the expression of downstream target genes, producing physiological changes in plants in response to HS. However, current research on the response of medicinal plants to HS is still in its infancy, primarily focusing on transcriptome and gene differential expression analysis, with functional studies of specific genes still limited. Meanwhile, researchers are exploring exogenous regulation to improve heat tolerance in medicinal plants, finding that applying substances like salicylic acid, melatonin, and calcium can effectively improve plant thermotolerance [8,9,10].

Identifying key molecular players involved in HS responses can lead to the development of molecular breeding techniques for improving heat tolerance in medicinal plants. In this review, we focus on recent findings of medicinal plants responding to HS at the physiological and molecular levels and summarize some representative preventive and control measures against heat damage.

## 2. Physiological Response of Medicinal Plants to HS

The physiological response of medicinal plants to HS involves various mechanisms, including growth morphology, photosynthesis, respiration, transpiration, and cell membrane stability, to help plants cope with high temperatures and minimize damage (Figure 1).

### 2.1. Growth Morphology

High temperatures can slow growth rates and affect the development of medicinal plant leaves. When plants are exposed to high temperatures for a short time, the leaves will curl and droop; if exposed for a long time, the plants will seriously wither and be close to death [11]. Simultaneously, plant height, weight, and shoot and leaf weight (both fresh and dry) decreased significantly under HS [4,12,13]. As a result, the production of medicinal plants is severely impaired after exposure to higher temperatures or extended periods of HS. However, proper heat treatment can promote the growth of some medicinal plants that prefer moisture and shade. *Houttuynia cordata* Thunb. (HC) plants survived at 30/25 °C and 35/30 °C treatments and exhibited significantly higher plant heights, leaf numbers, soil–plant analysis development, and normalized difference vegetation index values compared to lower temperature treatments [12].

### 2.2. Photosynthesis and Respiration

High temperatures can impair the photosynthetic machinery, including damage to the photosynthetic apparatus, inhibition of photosynthetic enzyme activity, and disruption of the photosystem II (PSII) complex, thereby reducing the efficiency of carbon fixation [11,14,15]. *Pinellia ternata* initially exhibited tolerance to the increased temperature, supported by typical growing leaves and sustained photosynthetic parameters [16]. However, when subjected to severe stress, *P. ternata* mesophyll cells were seriously damaged, with blurred chloroplast thylakoids, broken granular and stroma sheets, and accumulated granular thylakoids, resulting in a sharp decrease in the photosynthetic rate [16]. Furthermore, HS often leads to increased respiration rates, depleting the plant’s energy reserves and reducing growth [17].

### 2.3. Water Relations

Elevated temperatures can increase transpiration rates, leading to significant water loss compared to lower temperatures [18]. To conserve water, plants may close their stomata, which also limits CO₂ uptake and further reduces photosynthesis [19]. Two Himalayan herbs’ transpiration and stomatal conductance increased after HS at 35 °C [20]. Compared to heat-tolerant varieties of *Clematis*, heat-sensitive varieties showed earlier and sharply enhanced transpiration under HS, resulting in early water loss and wilting of the leaves [21]. In calendula cultivars, stomatal size increased significantly with longer durations of HS, and the heat-tolerant cultivar showed higher stomatal densities [22]. 

### 2.4. Membrane Sustainability

HS can compromise plant cell membrane integrity, leading to leakage of ions and other cellular contents, which affects overall cell function [15]. Electrolyte leakage (EL) and lipid peroxidation are important parameters indicating HS-induced plant membrane damage [23,24]. After exposure to HS, the relative electrolyte leakage (REL) and malondialdehyde (MDA) content of *P. ternata* leaf tissue increased following the treatments [25]. Long-term HS exposure significantly increased the EL and MDA contents in *Codonopsis tangshen* leaves, indicating that HS caused cell membrane damage in these plants [26]. Therefore, through the adaptation of lipid components and the enhancement of membrane lipid synthesis, the sustainability of cell membranes may be maintained, which is essential for resistance to HS.

These physiological responses collectively help medicinal plants withstand HS and maintain their physiological functions, growth, and productivity under high-temperature conditions. However, the specific response may vary among plant species and genotypes, highlighting the importance of understanding the mechanisms involved in HS tolerance in different medicinal plants.

## 3. Cellular Response of Medicinal Plants to HS

HS elicits various cellular responses in medicinal plants to mitigate damage and ensure survival. Intracellular responses to HS trigger the induction of relevant gene expression, the accumulation of heat shock proteins and metabolites, and the management of reactive oxygen species (Figure 1).

### 3.1. Heat Shock Proteins (HSPs)

Natural protein synthesis is blocked when plants are exposed to temperatures above the optimum. HS increases the accumulation of heat shock proteins (HSPs) [27]. These HSP proteins help stabilize and refold denatured proteins, protecting cells from heat-induced damage [28]. HSPs also act as molecular chaperones, ensuring proper protein folding and preventing the aggregation of misfolded proteins [28]. The transcript expression of some *HSPs* also showed up-regulated expression after HS treatments of *Lilium formolongi* [29]. In response to HS, the small HSPs of lily play an essential role in the crosstalk between HSF-HSP and ROS pathways, and in some specific tissues, this protects cells from extreme temperatures [30,31].

### 3.2. Reactive Oxygen Species (ROS) Management

HS leads to the production of reactive oxygen species in plant cells, which can damage cellular components [32,33]. For this reason, plants activate antioxidant enzymes like superoxide dismutase (SOD), catalase (CAT), and ascorbate peroxidase (APX) to scavenge ROS and mitigate oxidative stress [34]. However, in addition to toxic effects, ROS have been found to function as signaling molecules when plants respond to HS [35,36]. The content of ROS, such as H_2_O_2_ and O_2_^-^ in basil leaves, increased significantly with the prolongation of heat treatment time [11]. Comparing the heat tolerance of three *Coptis* species, it was found that *C. chinensis* showed activation of antioxidant enzymes, increased total antioxidant activity, and low ROS accumulation under HS conditions. These changes may be essential factors in maintaining the thermal acclimation and average growth of *C. chinensis* in low-altitude areas [37].

### 3.3. Metabolic Adjustments

Temperature significantly influences the production of metabolites in medicinal plants, including osmoprotectants and secondary metabolites (SMs) [38,39]. Accumulating compatible solutes such as proline, glycine betaine, and sugars helps in osmotic adjustment and protection of cellular structures [40,41]. HS can alter the production of SMs, which may enhance the plant’s defensive mechanisms or medicinal properties [42,43]. Combined with transcriptomic and metabolomic analyses, heat-tolerant *Lycium barbarum* L. species showed earlier expression and accumulation of amino acids and alkaloids associated with high temperatures and exhibited better heat tolerance [44]. In *Picrorhiza kurroa*, it was observed that some non-structural carbohydrates and amino acids (such as branched-chain amino acids and aromatic amino acids) were significantly accumulated in leaves under elevated temperatures. In contrast, SMs (such as picroside-I, picroside-II, picroside-III, feruloyl catalpol isomer, ferulic acid, and vanillic acid) increased in leaves but decreased in rhizomes, which may be due to the passive diffusion of SMs through plasmalemma at the soil-root interface [45].

### 3.4. Gene Expression

The activation of specific genes involved in the HS response includes those encoding *HSPs*, transcription factors, antioxidants, and metabolic enzymes. Modifications in DNA methylation, histone acetylation, and small RNAs can regulate the expression of stress-responsive genes [3,46]. In *Rubus idaeus* L., transcriptomics results revealed significant alterations in HSP family genes, SOD, peroxidase (POD), CAT, and photosynthesis-related differentially expressed genes (DEGs) under high-temperature stress, showing that these DEGs were mainly enriched in the pathways of photosynthesis-antenna proteins, pentose and glucuronide interconversion, phenylpropane biosynthesis, and indole alkaloid biosynthesis [14]. Comparative transcriptome analysis showed that 352 DEGs were upregulated in *Lilium longiflorum* (thermo-tolerant) and downregulated in *Lilium distichum* (thermo-sensitive) during HS, including 4-coumarate, CoA ligase, caffeoyl-CoA O-methyltransferase, peroxidase, pathogenesis-related protein 10 family genes, 14-3-3 protein, leucine-rich repeat receptor-like protein kinase, and glycine-rich cell wall structural protein-like. These genes were involved in metabolic pathways, phenylpropanoid biosynthesis, plant-pathogen interactions, plant hormone signal transduction, and kinase signaling pathways [47].

The cellular response of medicinal plants to HS involves a highly coordinated network of molecular and biochemical mechanisms to protect cellular integrity and function. These responses enable plants to cope with and adapt to high-temperature conditions, ensuring their survival and continued medicinal efficacy. Understanding these responses at the cellular level can help in developing strategies to enhance medicinal plants’ thermotolerance.

While the physiological and cellular responses to HS share many commonalities across various plant species, medicinal plants often exhibit unique adaptations or heightened sensitivities due to their specialized metabolic pathways and the production of bioactive compounds. Plant secondary metabolites are multifunctional metabolites that are associated with plant color, taste, scent, and response to stress [48]. Compared with non-medicinal plants, HS can significantly alter the synthesis and accumulation of characteristic secondary metabolites in some medicinal plants, such as alkaloids, flavonoids, and terpenoids, which are crucial for their medicinal properties [2,39,40]. Some medicinal plants may increase the production of specific stress-related secondary metabolites as a protective mechanism, while others may see a decline in these compounds (Figure 1). While non-medicinal plants also produce secondary metabolites, these compounds are often less critical to their overall fitness and economic value. Therefore, changes in secondary metabolite levels under HS may be less pronounced or impactful. Consequently, medicinal plants often exhibit more sensitive and finely tuned physiological and cellular responses to HS. These differences underscore the need for tailored approaches to managing HS for medicinal plants to ensure the preservation of their unique bioactive properties.

## 4. Molecular Mechanisms of Medicinal Plants for HS

Recent studies systematically summarize heat-responsive gene regulatory networks in *Arabidopsis* and crop plants [6]. The molecular mechanisms of plants in response to HS involve a complex network of signaling pathways, gene expression regulation, protein protection and turnover, and metabolic adjustments [6,43,46,49]. Due to the limitations of molecular techniques, research on the molecular mechanism of HS response in medicinal plants is still limited and needs improvement, mainly focusing on HSF, HSP, and transcription factors. Understanding these mechanisms is crucial for developing strategies to enhance the thermotolerance of medicinal plants, ensuring their survival and medicinal efficacy under heat stress conditions.

### 4.1. Molecular Responses of Lily to HS

Based on published studies, we summarize the molecular responses of lily to HS (Figure 2). Previous studies have shown that heat shock transcription factor A1 (HSFA1) plays a key role in the response to HS in plants, acting as a “master regulator” essential for transcriptional network activation [46]. Similarly, the lily (*Lilium longiflorum*) HS transcription factor LlHSFA1 interacts with LlHSFA2 and enhances heat tolerance in transgenic *Arabidopsis thaliana* overexpressing *LlHsfA1* by activating multiple downstream HS-related target genes [50]. The predicted target genes of LlHSFA1 include *HsfA2*, *HsfA7a*, *HsfA7b*, *DREB2A* (*DEHYDRATION-RESPONSIVE ELEMENT BINDING PROTEIN 2A)*, *MBF1c (MULTIPROTEIN BRIDGING FACTOR 1C)*, *GOLS1 (GALACTINOL SYNTHASE1)*, and some *HSPs* [50]. *LlDREB2B* is a homologous gene of *AtDREB2A* in *A. thaliana*, regulated at the post-translational level and controlling HS response in a DREB2B-HSFA3 module manner [51]. Furthermore, LlWRKY22 promotes thermotolerance through autoactivation and activation of *LlDREB2B* [52], while LlNAC014 increases thermotolerance by sensing high temperature and translocating to the nucleus to activate the DREB2-HSFA3 module (Figure 2) [53].

LlHSFA2 is a lily heat shock transcription factor that responds to HS and can also be regulated by the calcium signaling pathway [10,54]. LlHSFA2 interacts with LlHSFA1 and the ethylene response factor LlERF110 in the nucleus to co-regulate downstream target genes in response to HS [50,55]. Multiple transcription factors also regulate *LlHsfA2*. On the one hand, HD-Zip I transcription factor LlHB16 promotes heat resistance by activating *LlHsfA2* and *LlMBF1c* [56]. On the other hand, another HD-Zip I protein, LlHOX6, interacts with LlHB16 to limit its transactivation and impair the HS response of lily (Figure 2) [57]. MBF1c is a highly conserved transcriptional coactivator that plays a vital role in the HS response [58,59]. Similar to *LlHsfA2*, *LlMBF1c* is regulated by both LlHOX6 and LlHB16 [56,57]. In addition, LlWRKY39 is also an upstream regulator of *LlMBF1c*, and LlCaM3 has a negative effect on the role of LlWRKY39 in the transcriptional activation of *LlMBF1c*, forming a feedback regulation pathway to balance the LlWRKY39-mediated HS response (Figure 2) [60].

During plant response to HS, HSF transcription factors usually exert regulatory effects on each other in a manner that forms homo- or heterologous interactions. Two lily homologous *HsfA3* (heat stress transcription factor A3) genes, *LlHsfA3A* and *LlHsfA3B*, overexpressed in *Arabidopsis*, enhance heat tolerance in transgenic plants, which may implicate proline catabolism [61]. LlNAC014 can bind the CTT(N7)AAG element of the *LlHsfA3A* and *LlHsfA3B* promoters to activate their expression [53]. LlERF110 could also directly bind to the *LlHsfA3A* promoter and activate its expression [55]. Furthermore, the heat-inducible splice variant LlHSFA3B-III interacts with LlHSFA3A-I to limit its transactivation function, reducing the adverse effects of excessive LlHSFA3A-I accumulation in short-term HS response. LlHSFA3A-I can also activate the transcription of *LlHSFA3B* to form a regulatory loop, while LlHSFA3B-III antagonizes LlHSFA3A-I to repress the expression of LlHSFA3A-I-induced genes and positively regulates the long-term HS response (Figure 2) [62]. The lily heat-inducible HSFC2 homolog, LlHSFC2, plays an active role in the overall homeostasis and maintenance of the HS response by cooperating with HSFAs [63]. LlHSFC2 can interact with itself and multiple HSFAs. LlHSFC2 interacts with HSFAs to accelerate their transcriptional activation capacity and acts as a transcriptional coactivator. After exposure to HS, the homologous interaction of LlHSFC2 is repressed. Still, its heterologous interaction with heat-inducible HSFAs is promoted, allowing it to play a coactivating role in establishing and maintaining heat tolerance (Figure 2) [63].

HS induces a large amount of ROS in chloroplasts and mitochondria, and excessive ROS can cause damage to plants [64]. Therefore, plants produce ROS-scavenging systems to protect cells. Another HS transcription factor, LlHSFA4, enhances essential heat tolerance in lily by regulating ROS metabolism [65]. Heterologous expression of David lily (*Lilium davidii* (E. H. Wilson) Raffill var. Willmottiae) *LimHSP16.45* in *Arabidopsis* prevents irreversible protein aggregation and scavenges reactive oxygen species from the cell, protecting the plant from HS (Figure 2) [66]. Additionally, the R2R3-MYB transcription factor LlMYB305 in lily plays a positive role in heat tolerance by activating the heat-protective gene *LlHSC70*, a homolog of HSP70s (Figure 2) [67]. The key response transcription factors induced by HS, such as HSFs and MBF1c, further bind to downstream heat-responsive gene promoters and activate their expression to enhance plant thermotolerance.

### 4.2. Molecular Responses of Other Medicinal Plants to HS

Compared with lily, the molecular mechanisms of other medicinal plants in response to HS are more superficial. Several *HSP* and *HSF* genes have been identified and preliminarily functionally characterized, such as *PtHSP18.2* [68], *PtsHSP17.2* [69], *ClHSP20s* [70], and *SmHSFs* [71]. The expression of *PtHSP18.2* and *PtsHSP17.2* increased under HS in *Pinellia ternata*. *PtHSP18.2*, when overexpressed in *Escherichia coli*, showed an association with the maintenance of cell viability under HS, whereas *PtsHSP17.2*, when overexpressed in tobacco, showed a significant increase in heat tolerance in the transgenic plants [68,69]. Genome-wide identification, evolutionary analysis, sequence characterization, and expression analysis of the *ClHSP20* and *SmHSF* gene families were performed in the *Coix lacryma-jobi* L. and *Salvia miltiorrhiza* genomes [70,71].

Several other genes identified in medicinal plants are involved in response to HS, including enzymes, transcription factors, and 14-3-3 proteins. Under HS and cold stress, the expression of three sterol glycosyltransferase (SGT) genes increased in *Withania somnifera*, and one of the SGT enzymes was glycosylated with stigmasterol, indicating the role of sterol modification in abiotic stress [72]. In *S. miltiorrhiza*, three galactoside synthase genes (*SmGols*) have been shown to have significant functions in response to cold or heat, which appear to be regulated by several HSF transcription factors [73]. Also, the three *SmUSPs* (*SmUSP1*, *SmUSP8,* and *SmUSP27*) expressed in *E.coli* showed stronger tolerance to salt, heat, and a combination of the two, indicating that these proteins have protective effects when cells are exposed to single and multiple stress conditions [74]. 62 WRKY family genes were identified and characterized from the *Dendrobium catenatum* genome, among which *DcWRKY22* was highly induced in roots subjected to heat and may act in an ABA- and SA-dependent manner in response to HS [75]. Similarly, forty-two 14-3-3 genes were identified from the ginseng genome, and the qRT-PCR results indicated multiple expression patterns of 14-3-3 genes under high-temperature stress, implying that multiple 14-3-3 genes were involved in the high-temperature stress response [76].

Although many genes involved in HS response and their regulatory mechanisms have been identified in *A. thaliana* and crops, most studies on the molecular mechanisms of heat response in medicinal plants are still limited to transcriptome sequencing and differential analyses, with only a small number of genes investigated in more depth.

## 5. Exogenous Regulation of Enhancing Thermotolerance in Medicinal Plants

Although higher plants develop their own defense strategies to overcome the effects of HS, these are often insufficient, and some strategies, such as mass production of ROS, may also cause significant damage. Numerous studies have demonstrated that exogenously applied phytohormones, chemicals, signaling molecules, and other substances can be protective agents to confer high-temperature stress tolerance to plants [77]. Similarly, some exogenous substances have been reported to improve the heat resistance of medicinal plants (Table 1).

Salicylic acid (SA) is an essential molecule in the signal transduction pathway of biotic and abiotic defense responses, contributing to plant resistance [84,85]. As a plant growth regulator, exogenous application of SA can induce plant heat tolerance [86]. Several medicinal plants have demonstrated that exogenous application of SA improves plant thermotolerance, such as *Digitalis trojana* [78], *Mentha-piperita* and *Mentha arvensis* [8], *Salvia officinalis* and *Salvia elegans* [9], and *Origanum vulgare* [79]. SA treatment significantly increased the activity of antioxidant enzymes and the total phenol and proline levels in medicinal plants, which may play an essential role in high-temperature stress resistance [8,78,79]. Additionally, SA may promote the production of critical active ingredients in medicinal plants. For example, in *D.troana*, after pretreatment of the callus with SA and exposure to high temperatures for two hours, a significant increase in the production of cardenolides and the ability to induce heat tolerance was observed [79]. Combining SA and other effective substances can also improve the heat tolerance of medicinal plants. Thus, the combined treatments of 100 μM SA and 5 mM CaCl_2_ in *Salvia* and 30 mM melatonin and 4 mM SA in *Mentha*, respectively, were effective in alleviating the ability of the plants to cope with HS [8,9].

Exogenous addition of signaling factor substances is also a means of mitigating heat damage in medicinal plants. In *L. longiflorum*, exogenous ethylene and CaCl_2_ treatments enhanced heat resistance by triggering the ethylene signaling pathway and the Ca signaling pathway in response to HS, respectively [10,55]. Recent studies have found that exogenously applied calcium can regulate the antioxidant system by promoting flavonoid synthesis to alleviate HS in *Rhododendron* plants [81]. Exogenous growth regulator pretreatment reduces the damage caused by HS and is an effective method to improve heat tolerance in plants [87,88]. In *Mentha*, the application of melatonin alone can also alleviate the effect of HS [8]. Transcriptome analysis showed that exogenous application of melatonin and spermidine could improve the heat resistance of *P. ternata* by regulating a series of short-term HS response genes. However, they may have different regulatory patterns [80].

Some amino acids, such as proline, leucine, isoleucine, and valine, can regulate plant stress resistance by scavenging ROS or influencing the plant’s respiratory system [34,89]. In *Panax notoginseng*, exogenous foliar leucine enhances carbohydrate metabolism in multiple tissues of plants, including sugars and TCA cycle metabolites, to improve the energy supply of plants and further alleviate HS [83]. Alternatively, antimicrobial analogs may help plants enhance abiotic stress resistance. Trichokonins (TKs) are antimicrobials extracted from *Trichoderma longibrachiatum* strain SMF2 [90]. TKs highly induce the HSF-HSP pathway under HS conditions in Lanzhou lily, thereby enhancing heat tolerance; moreover, LzHsfA2a-1 may play a key role in acquiring heat tolerance induced by TKs [82].

In addition to the exogenous regulation measures mentioned above, some methods to improve heat tolerance can be applied to other plants, which can also be tried on medicinal plants. For example, endophytes and plant growth-promoting rhizobacteria (PGRR) are essential in mitigating HS through complex interactions with plants. Their interactions promote antioxidant activity and the accumulation of suitable osmotic solutes (such as proline, glycine betaine, soluble sugars, and trehalose), improving stressed plants’ nutritional status [91,92]. In *Rhododendron hainanense*, root-endophytic bacteria affect the plant’s ability to absorb nitrogen, and root-endophytic fungi control plant physiological traits by altering proline content, thereby increasing the resistance of *R. hainanense* to HS [93]. Furthermore, plant nanobiotechnology is a promising technology for sustainable agricultural production, promoting plant growth and resistance to biotic and abiotic stresses [94,95]. Nanoparticles can enhance the ability of plants to scavenge ROS, retain water, improve photosynthesis, and strengthen cell walls, aiding in drought and HS tolerance [96]. The application of nanomaterials in agriculture presents a promising avenue for enhancing plant resistance to various biotic and abiotic stresses. However, whether used for medicinal plants or other crops, it is essential to continue researching the environmental impact and safety of nanomaterial use in agriculture to ensure their responsible and practical application.

## 6. Summary and Perspectives

The intensification of global warming and the frequent occurrence of extreme temperatures significantly affect the growth and accumulation of active ingredients in medicinal plants. Understanding the physiological, biochemical, and molecular responses of these plants to HS and implementing a combination of breeding, agronomic, biostimulant, and advanced technological strategies can mitigate these effects and ensure the continued availability and efficacy of medicinal plants in a changing climate.

This review summarized medicinal plants’ physiological and cellular responses to HS. We discussed medicinal plants’ unique adaptability and high sensitivity to HS due to their particular metabolic pathways and accumulation of bioactive compounds. With the continuous development of sequencing technology, the study of the molecular genetic information of medicinal plants has been facilitated [97]. Therefore, we took lily as an example to summarize the molecular response mechanism of medicinal plants to HS, mainly involving the response and regulation of transcription factors such as HSF, ERF, WRKY, MYB, and NAC (Figure 2). We also collected some molecular regulation mechanisms reported in other medicinal plants. Most medicinal plants are limited in the depth of molecular studies due to their unique habitats and complex genomes [98,99]. With the rapid development of molecular biology and biotechnology, transcriptomics, proteomics, and multi-omics interlinked analyses, the mechanism of medicinal plant response to HS is expected to be studied in greater depth. Finally, we summarized some exogenous control measures that can improve the thermotolerance of medicinal plants (Table 1). Exogenous application of plant hormones, plant growth regulators, signal factors, amino acids, and other substances can effectively promote plant heat resistance. Of course, to ensure the safety of medicinal functions, some other heat-resistant prevention and control measures, such as beneficial microorganisms and nanomaterials, can also be tried on medicinal plants to resist high temperatures.

Future research on heat tolerance in medicinal plants will mainly focus on several key aspects. First, exploring the regulatory roles of heat stress-responsive genes and transcription factors in medicinal plants in response to heat stress is crucial. Second, constructing comprehensive networks of signaling pathways for HS perception in plants will be essential. Third, investigating the molecular mechanisms by which HS affects secondary metabolic pathways in medicinal plants. Fourth, developing strategies to rationally utilize exogenous regulatory measures to simultaneously improve heat tolerance and the accumulation of secondary metabolites of active ingredients in medicinal plants is vital. The rational use of modern molecular biology technologies, combined with multi-omics approaches, molecular-assisted breeding, gene editing, and nanomaterials, offers a powerful framework for developing heat-tolerant medicinal plant varieties. By integrating these technologies in a multidisciplinary framework, it is possible to enhance the resilience of medicinal plants to heat stress, ensuring sustainable production and maintaining the quality of their active ingredients.

## Figures and Tables

**Figure 1 ijms-25-08600-f001:**
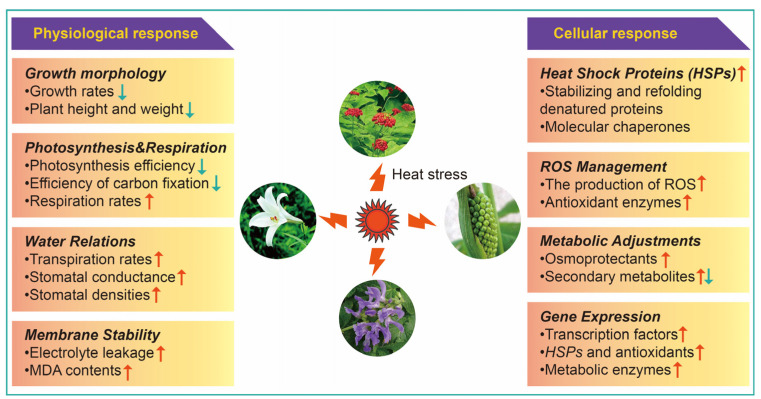
Physiological and cellular response of medicinal plants to heat stress (HS). Upward-red pointing arrows indicate activated or increased response indicators, while downward-green pointing arrows indicate inhibited or decreased response indicators—abbreviations: MDA, malondialdehyde; ROS, reactive oxygen species.

**Figure 2 ijms-25-08600-f002:**
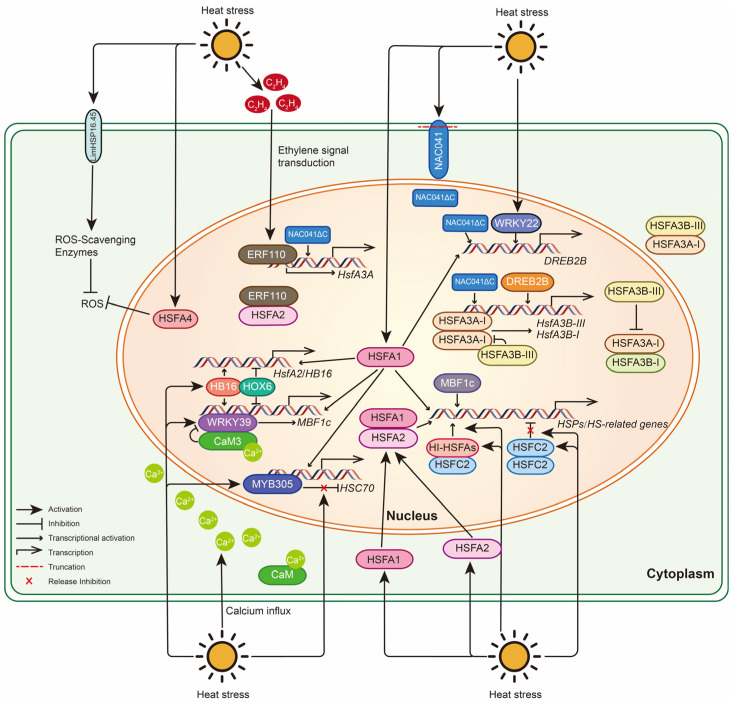
Molecular responses to Heat Stress (HS) in lily. Under HS, multiple types of transcription factor genes are activated, including HSF, WRKY, MYB, NAC, and ERF family genes. HSFA1, a core heat-responsive transcription factor, interacts with HSFA2 to activate the expression of multiple downstream target genes. The NAC041 protein is activated to truncate its C-terminus, prompting NAC041∆C to enter the nucleus and function as a transcription factor. NAC041∆C further targets binding to *DREB2B* and *HsfA3s* promoters to activate their expression. *LlHsfA3B* undergoes heat-inducible alternative splicing to produce LlHSFA3B-III, which limits LlHSFA3A-I function in short-term HS response by interfering with the formation of functional oligomers. LlHSFA3A-I also activates the transcription of *LlHsfA3B* to form a regulatory loop to inhibit the adverse effects of strong expression of *LlHsfA3A*. Additionally, ethylene accumulation induces the expression of *ERF110*, which directly binds to the promoter of *HsfA3A* to regulate its expression and interacts with HSFA2 to mediate the expression of HS-responsive genes. WRKY39 is rapidly induced, directly activating the expression of *MBF1c*; simultaneously, Ca^2+^ enters the cell to activate CaM3, which interacts with WRKY39 to prevent the side effects of excessive activation. HB16 promotes thermotolerance by activating *HSFA2* and *MBF1c*, whereas HOX6 interacts with HB16 to limit its transcriptional activation. The accumulation of *HSFC2* and *HSFA* transcripts increases, and the homologous interaction of HSFC2 weakens, making it tend to form oligomers with heat-induced HSFA (HI-HSFA) members, thereby forming transcriptional coactivators and enhancing the induction of downstream heat-responsive genes. MYB305 represses the promoter activity of *LlHSC70* under normal conditions but activates it under HS. HSFA4 and membrane-localized LimHSP16.45 protect plant cells by regulating the scavenging of reactive oxygen species.

**Table 1 ijms-25-08600-t001:** Representative studies on exogenous regulation measures for Heat Stress (HS) in medicinal plants.

Species	Applied Technical Measures	Mode of Application	Mechanism of Heat Resistance	Reference
*Digitalis trojana* Ivanina	150 mM Salicylic acid	Culture medium	Induced synthesis of antioxidants and cardenolides	[78]
*Mentha-piperita* L. (Mitcham variety) and *Mentha arvensis* L. (var. *piperascens*)	30 M Melatonin, 4 mM Salicylic acid	Foliar spray	Increased antioxidant enzyme activity	[8]
*Origanum vulgare* L.	1 mM Salicylic acid	Foliar spray	A rise in the activity of superoxide dismutase and the levels of total phenol and hydrogen peroxide	[79]
*Salvia officinalis* L. and *Salvia elegans* Vahl	100 μM Salicylic acid and 5 mM CaCl_2_	Soil watering	Increasing values of soil–plant analysis development, normalized difference vegetation index, and the maximal quantum yield of photosystem II photochemistry	[9]
*Pinellia ternata*	100 μM Spermidine, 100 μM Melatonin	Foliar spray	Up-regulation of heat-responsive genes	[80]
*Rhododendron × pulchrum*	10 mM CaCl_2_	Foliar spray	Inducing the production of flavonoid compounds to regulate the antioxidant system	[81]
*Lilium longiflorum*	20 mM CaCl_2_	Apical treatment	LlCaM3 is a major component in Ca^2+^-CaM HS signaling pathway in lily and might be in the upstream of HSF	[10]
*Lilium longiflorum*	2 ppm Ethylene	Seedling treatment	LlERF110 mediates HS response via regulation of LlHsfA3A expression and interaction with LlHsfA2	[55]
*Lilium davidii* var. *unicolor*	0.5, 1, 2, 4, or 8 mg/L Trichokonins	Root treatment	Inducing the HSF-HSP pathway and LzHsfA2a-1 likely plays a key role in acquisition of TKs-induced thermotolerance	[82]
*Panax notoginseng*	3 and 5 mM leucine	Foliar spray	Enhanced the antioxidant capacity, carbohydrate metabolism, and TCA cycle metabolites	[83]

## Data Availability

Not applicable.

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
