# Peer review of "Research Progress on Heat Stress Response Mechanism and Control Measures in Medicinal Plants"

_ijms, 2024, doi:10.3390/ijms25168600_

Round 1

Reviewer 1 Report

Comments and Suggestions for Authors

Dear authors,

This review article will be helpful to the researchers who focus on medicinal plants or heat stress.

However, there were some insufficient explanation.

Please check points in attached file.

Comments on the Quality of English Language

Dear Editor,

I'm sorry for being late review.

I would like to inform you that this paper is acceptable after revision of minor points.

Thank you for your efforts.

Sincerely yours,

Reviewer 

Author Response

Comments 1: What kind of specialize habitats? Please explain it detail.
Response 1: Thanks for your suggestion. P1, Line 27-29, as suggested by the reviewer, we have added a sentence to explain the "specialized habitats". 

Comments 2: Describe more. What kind of deleterious effects in plants?
Response 2: P1, Line 41-43, as suggested by the reviewer, we have added content to the description of deleterious effects. 

Comments 3: Insufficient explanation for the mechanisms by HS, and how HSFs affect the HS. In addition, it recommended that author may add the lack of study related to the mechanism by HS.

Response 3: Thanks for your suggestion. P2, Line 49-52, as suggested by the reviewer, we have added the elaboration of  how HSFs affect the HS and added a reference. The literature is as follows:
Andrási, N.; Pettkó-Szandtner, A.; Szabados, L., Diversity of plant heat shock factors: regulation, interactions, and functions. J. Exp. Bot. 2020, 72, 1558-1575.

Comments 4: It recommended that author may emphasize the limited study about specific genes.
Response 4: Thanks for your suggestion. Here is a summarized description, while some specific genes of the study limitations are mainly detailed in section 4.2.

Comments 5: Please describe detail. Whether it heat-tolerant plants or not.
Response 5: Thank you for pointing this out. P3, Line 85, we have added a detailed description of medicinal plants.

Comments 6: Author should add more detail relationship between the photosynthesis and HS.
Response 6: Thanks for your suggestion. P3, Line 90-92, we have added a detailed description between the photosynthesis and HS.

Comments 7: Add the way to maintain cell membrane in plants including HS.
Response 7: P3, Line 117-119, we have added the ways to maintain cell membrane in plants including HS.

Comments 8: Add detailed explanation the reason for decreased SMs in rhizomes.
Response 8: P4, Line 165-166, we have added details explaining the reasons for the decrease in SMs in rhizomes.

Comments 9: Please describe detail. Whether it heat-tolerant plants or not.
Response 9: Thank you for pointing this out. P5, Line 193-196, based on the reviewer 's suggestion, we have added and amended the description of non-medicinal and medicinal plants here to make the content more rigorous. The added literature is as follows:
Jan, R.; Asaf, S.; Numan, M.; Lubna; Kim, K.-M., Plant Secondary Metabolite Biosynthesis and Transcriptional Regulation in Response to Biotic and Abiotic Stress Conditions. Agronomy. 2021, 11, 968.

Reviewer 2 Report

Comments and Suggestions for Authors

This review highlights various strategies to enhance heat tolerance in medicinal plants through exogenous applications and biotechnological methods. The research underscores the importance of salicylic acid (SA), melatonin, calcium chloride (CaCl2), and other substances in improving thermotolerance and promoting the accumulation of beneficial compounds in plants. The reviewed studies provide a comprehensive understanding of how various exogenous applications and biotechnological interventions can enhance the heat tolerance of medicinal plants. These approaches not only help in sustaining plant growth under high temperatures but also ensure the quality and efficacy of their active ingredients. Future research should focus on molecular mechanisms, signaling pathways, and the safe application of nanomaterials to further improve thermotolerance in medicinal plants.

Overall, the content of this review is unique, and the authors have prepared an excellent manuscript. I have the following minor comments on the manuscript:

In Figure 1, please use different colors to indicate activation and deactivation of various responses to heat stress.

Moreover, the quality of Figures 1 and 2 should be improved. Authors should provide high-quality figures.

Author Response

Comments 1: In Figure 1, please use different colors to indicate activation and deactivation of various responses to heat stress.

Response 1: Thanks for your suggestion. We have changed the color of downward-pointing arrows to green, which indicates inhibited or decreased response indicators. P2, Line 73-74, we also have changed the corresponding legend.

Comments 2: Moreover, the quality of Figures 1 and 2 should be improved. Authors should provide high-quality figures.

Response 2: Thanks for your suggestion. In order to improve the quality of the image, we have changed the pixels of Figures 1 and 2 to 600dpi.